# Review of Therapeutic Strategies for Anaplastic Lymphoma Kinase-Rearranged Non-Small Cell Lung Cancer

**DOI:** 10.3390/cancers14051184

**Published:** 2022-02-24

**Authors:** Takafumi Fukui, Motoko Tachihara, Tatsuya Nagano, Kazuyuki Kobayashi

**Affiliations:** Division of Respiratory Medicine, Department of Internal Medicine, Kobe University Graduate School of Medicine, Kobe 650-0017, Hyogo, Japan; takafumi_04garnet91@icloud.com (T.F.); tnagano@med.kobe-u.ac.jp (T.N.); kkoba@med.kobe-u.ac.jp (K.K.)

**Keywords:** non-small-cell lung cancer (NSCLC), anaplastic lymphoma kinase (ALK), tyrosine kinase inhibitors (TKI), angiogenesis inhibitors, immune checkpoint inhibitor (IO), resistance mechanism, biopsy

## Abstract

**Simple Summary:**

Anaplastic lymphoma kinase (ALK)-rearranged non-small cell lung cancer (NSCLC) was first reported in 2007. Following the development of crizotinib as a tyrosine kinase inhibitor (TKI) targeting ALK, the treatment of advanced NSCLC with ALK-rearrangements has made remarkable progress. Currently, there are five ALK-TKIs approved by the FDA, and the development of new agents, including fourth-generation TKI, is ongoing. Clinical trials with angiogenesis inhibitors and immune checkpoint inhibitors are also underway, and further progress in the treatment of ALK-rearranged advanced NSCLC is expected. The purpose of this manuscript is to provide information on the recent clinical trials of ALK-TKIs, angiogenesis inhibitors, immune checkpoint inhibitors, and chemotherapy, to describe tissue and liquid biopsy as a method to investigate the mechanisms of resistance against ALK-TKIs and suggest a proposed treatment algorithm.

**Abstract:**

Non-small cell lung cancer (NSCLC) with anaplastic lymphoma kinase rearrangement (ALK) was first reported in 2007. ALK-rearranged NSCLC accounts for about 3–8% of NSCLC. The first-line therapy for ALK-rearranged advanced NSCLC is tyrosine kinase inhibitors (TKI) targeting ALK. Following the development of crizotinib, the first ALK-TKI, patient prognosis has been greatly improved. Currently, five TKIs are approved by the FDA. In addition, clinical trials of the novel TKI, ensartinib, and fourth-generation ALK-TKI for compound ALK mutation are ongoing. Treatment with angiogenesis inhibitors and immune checkpoint inhibitors is also being studied. However, as the disease progresses, cancers tend to develop resistance mechanisms. In addition to ALK mutations, other mechanisms, including the activation of bypass signaling pathways and histological transformation, cause resistance, and the identification of these mechanisms is important in selecting subsequent therapy. Studies on tissue and liquid biopsy have been reported and are expected to be useful tools for identifying resistance mechanisms. The purpose of this manuscript is to provide information on the recent clinical trials of ALK-TKIs, angiogenesis inhibitors, immune checkpoint inhibitors, and chemotherapy to describe tissue and liquid biopsy as a method to investigate the mechanisms of resistance against ALK-TKIs and suggest a proposed treatment algorithm.

## 1. Introduction

Anaplastic lymphoma kinase (ALK) was first discovered as a fusion partner in the (2;5) chromosomal translocation in anaplastic large cell lymphoma in 1994 by Morris et al. [1]. ALK is a transmembrane tyrosine kinase encoded by the ALK gene localized on chromosome 2, belonging to the superfamily of insulin receptors. ALK expression is considered to be involved in the development of the nervous system [2]. ALK regulates several pathways involved in cell survival, proliferation, and cell cycling, including the AKT/PI3K [3] and STAT3 pathways [4,5]. The echinoderm microtubule-associated protein-like-4 (EML4)-ALK fusion gene was discovered in non-small cell lung cancer (NSCLC) in 2007 [6]. The EML-ALK fusion gene results from the fusion of the EML-4 gene and the ALK gene by inversion within chromosome 2p. The EML4-ALK fusion protein, produced from the EML4-ALK fusion gene, promotes carcinogenesis by activating an intrinsic tyrosine kinase. The anti-tumor effect of ALK-TKIs results from binding to the ATP-binding site of the intracellular tyrosine kinase domain, regulating signal transduction. Driver mutation is identified in 70–80% of NSCLC [7]. ALK-rearrangement is associated with approximately 3–8% of NSCLC [8,9]. The patients reported to be likely to harbor ALK-rearrangement are young, never/light smokers with adenocarcinoma and have a higher risk of metastasis to the central nervous system [10,11]. In recent years, treatment for ALK-rearranged NSCLC has made remarkable progress. Nowadays, ALK lung cancer can be expected to survive 5 years. Currently, five ALK-TKIs are approved by the FDA. Since the FDA approval of crizotinib in 2011, ceritinib was approved in 2014, alectinib in 2015, brigatinib in 2017, and lorlatinib in 2018 (Figure 1). Furthermore, clinical trials of the novel ALK-TKIs, including ensartinib and fourth-generation ALK-TKI for compound ALK mutation, are ongoing. In addition, clinical trials of angiogenesis inhibitors, immune checkpoint inhibitors, and combination therapy of platinum doublet chemotherapy and ALK-TKI are ongoing. In this review, we provide information on the recent clinical trials on ALK-TKIs, angiogenesis inhibitors, immune checkpoint inhibitors, and chemotherapy, describe tissue and liquid biopsy as a method to investigate the mechanisms of resistance against ALK-TKIs, and suggest a proposed treatment algorithm.

## 2. Diagnosis

International guidelines recommend testing for ALK mutations in all non-squamous NSCLC [12,13]. There are several methods to detect ALK-rearrangement: fluorescence in situ hybridization (FISH), immunohistochemistry (IHC), and polymerase chain reaction (PCR)-based next-generation sequencing (NGS). The choice of which test to use, and the algorithm is controversial and is an issue for consideration. The advantage of NGS is that it can be used in liquid biopsy using circulating tumor cells (ctDNA), etc., not only tissue samples, and it can detect drug resistance mechanisms. Identification of the resistance mechanism to ALK-TKIs and corresponding treatment are expected to improve prognoses. Liquid biopsy is aimed at predicting drug resistance mechanisms and treatment efficacy using specimens such as ctDNA, cell-free DNA (cfDNA), and circulating tumor DNA in the blood. The validity of ALK-rearrangement detection by liquid biopsy using NGS was demonstrated in the Blood First Assay Screening Trial (BFAST) [14] and the NILE (Noninvasive versus Invasive Lung Evaluation) study [15]. In the BFAST study, alectinib was shown to have a non-inferior outcome in patients with ALK-rearranged NSCLC diagnosed by blood-based NGS compared to those diagnosed by tissue-based NGS. The NILE study showed that cfDNA analysis using Guardant360 was non-inferior to tissue samples in detecting guideline-recommended biomarkers. These results demonstrated that for patients who, for some reason, do not have ALK-rearrangement identified in tissue samples, there is an opportunity to treat with ALK-TKIs based on blood-based NGS results. However, it has been reported that the level of detection of ALK-rearrangements by liquid biopsy is lower than that by tissue biopsy due to the small amount of acid released into the blood in small or slow-growing tumors, etc. [16]. Tissue biopsy and liquid biopsy have their own advantages and disadvantages. Tissue rebiopsy has advantages in histological evaluation, allowing for the evaluation of EMT transformation and gene amplification. Biopsies, however, are highly invasive and expensive because of the need for bronchoscopy or surgery. Liquid biopsy is not capable of histological evaluation but is non-invasive and is therefore repeatable and cost-effective. A liquid biopsy may be particularly useful when tissue samples are insufficient or when tissue biopsy is contraindicated. Detecting driver mutations and ALK resistance mutations as therapeutic targets is very beneficial, even when sufficient tissue samples are not available for biopsy. In addition, liquid biopsy is easily repeatable, allowing for real-time, long-term monitoring during treatment and the early detection of recurrence before clinical symptoms appear [17]. Although liquid biopsy for ALK-rearranged NSCLC is still under investigation, it is expected to be a useful tool for detecting genetic mutations with false-negative results by tissue biopsy and investigating resistance mechanisms.

## 3. ALK-TKIs

### 3.1. Crizotinib

Crizotinib is a multi-kinase inhibitor of tyrosine kinases, such as ALK, c-MET, and ROS-1. Crizotinib was the first ALK-TKI approved for the treatment of ALK-rearranged NSCLC. In part 2 of the phase I PROFILE 1001 trial for ROS1-rearranged NSCLC patients, two patients with ALK-rearrangements who received crizotinib 250 mg orally twice daily showed a significant response. Therefore, the expanded cohort of patients with ALK-rearranged NSCLC stage III/IV was added in 2008 [18]. In this study, 143 patients with pre-treated or untreated ALK-rearranged NSCLC were included. Three patients had complete response (CR) and 84 patients had partial response (PR), with an overall response rate (ORR) of 60.8% (95% CI: 52.3–68.9%), median duration of response (DOR) of 49.1 weeks (95% CI: 39.3–75.4%), and median progression-free survival (mPFS) of 9.7 months (95% CI: 7.7–12.8%) [19]. The phase II PROFILE 1005 study included 1069 patients with previously treated ALK-rearranged advanced NSCLC [20]. In 908 patients determined to be ALK-positive at the central laboratory, the reported ORR was 54% (95% CI: 51–57), mPFS was 8.4 months (95% CI: 7.1–9.7), and median overall survival (mOS) was 21.8 months (95% CI: 9.4–24.0). Based on the results of phase I and II studies, in 2011, the FDA approved crizotinib as the first ALK-TKI for ALK-rearranged NSCLC. The phase III PROFILE 1007 study compared crizotinib with chemotherapy as second-line treatment in ALK-rearranged NSCLC patients whose disease had progressed after one prior platinum-based chemotherapy regimen [21]. The primary endpoint, mPFS, was 7.7 months (95% CI: 6.0–8.8) in the crizotinib arm (*n* = 173) and 3.0 months (95% CI: 2.6–4.3) in the chemotherapy arm (pemetrexed: 101, docetaxel: 73). ORR was 65% (95% CI: 58–72) with crizotinib compared to 20% (95% CI: 14–26) with chemotherapy (*p* < 0.001). Median DOR was 32.1 weeks (95% CI: 2.1–72.4) in the crizotinib arm and 24.4 weeks (95% CI: 3.0–43.6) in the chemotherapy arm (HR: 0.49, 95% CI: 0.37–0.64). Based on the results of the PROFILE 1007 study, crizotinib was approved in 2013 for the second-line treatment of patients with ALK-rearranged NSCLC whose disease has progressed after platinum-based chemotherapy. In 2014, the phase III PROFILE 1014 study was conducted to evaluate the efficacy of crizotinib compared to pemetrexed plus platinum doublet chemotherapy as a first-line treatment for patients with ALK-rearranged advanced NSCLC [22,23]. In the PROFILE1014 study, the primary endpoint, mPFS, was 10.9 months (95% CI: 8.3–13.9) in the crizotinib arm (*n* = 172) and 7.0 months (95% CI: 6.8–8.2) in the chemotherapy arm (*n* = 171) (HR: 0.45 (95% CI: 0.35–0.60)). ORR was 74% (95% CI: 67–81) in the crizotinib arm and 45% (95% CI: 37–53) in the chemotherapy arm (*p* < 0.001). The median DOR was 11.3 months (95% CI: 8.1–13.8) and 5.3 months (95% CI: 4.1–5.8), respectively. Based on the result, crizotinib was approved as a first-line treatment for ALK-rearranged advanced NSCLC. In the recent analysis of the PROFILE 1014 study, mOS was not reached (NR) (95% CI: 45.8–NR) in the crizotinib arm and 47.5 months (95% CI: 32.2–NR) in the chemotherapy arm with HR of 0.76 (95% CI: 0.548–1.053, *p* = 0.0978), which was not statistically significant. Despite the prolonged PFS, crizotinib did not significantly improve OS, which may be due to the fact that 84.2% of patients in the chemotherapy arm crossed over to crizotinib and more effective treatments, such as other ALT-TKIs or chemotherapy, as subsequent therapy may have affected the results.

#### 3.1.1. Intracranial Efficacy

It has been reported that crizotinib has low penetration of the blood–brain barrier (BBB) [24,25]. Lower cerebrospinal fluid (CSF) concentrations and CSF/plasma ratios prevent the achievement of therapeutic concentrations in the brain and lead to pharmacological tolerance. In the PROFILE 1014 study, the incidence of extracranial PD only was less with crizotinib than with chemotherapy, regardless of the presence or absence of brain metastasis (BM) prior to treatment initiation (ITT population: 73% vs. 80%, BM present: 57% vs. 60%, BM absent: 78% vs. 86%). In contrast, the proportion of patients for whom the brain was the only site of PD was higher with crizotinib than with chemotherapy (ITT population: 24% vs. 10%, BM present: 38% vs. 23%, BM absent: 19% vs. 6%). In contrast to crizotinib, the next-generation ALK-TKIs passed through the BBB and had higher concentrations in the CSF. The clinical trial comparing crizotinib with the next-generation ALK-TKIs showed that crizotinib had inferior activity in the CNS.

#### 3.1.2. Safety

The characteristic adverse events of crizotinib are visual disturbances (diplopia, photophobia, and blurred vision). A total of 73% of patients in the PROFILE 1014 study had visual disturbances. The other frequent adverse events were gastrointestinal disorders: diarrhea occurred in 66% and nausea in 59% of patients. The most frequent grade 3–4 adverse events were neutropenia (15%), elevated transaminases (14%), and pulmonary embolism (8%). In the PROFILE 1014 study, there was no significant difference in severe (RR: 0.97, 95% CI: 0.79–1.18) and fatal (RR: 2.24, 95% CI: 0.49–10.30) adverse events between crizotinib and chemotherapy.

Recently, crizotinib has been used less frequently due to its inferior intracranial efficacy and shorter mPFS compared to next-generation ALK-TKIs.

### 3.2. Ceritinib

Ceritinib was approved by the FDA in 2014 based on the results of ASCEND-1 [26] and ASCEND-2 [27] for the treatment of NSCLC patients with ALK-rearrangement NSCLC who have progressed during treatment with crizotinib or who cannot tolerate crizotinib. In the phase I ASCEND-1 trial, 255 ALK-rearranged advanced NSCLC patients were treated with 750 mg/day of ceritinib. In the ALK-TKI naive cohort (*n* = 83), ORR was 72.3% (95% CI: 61.4–81.6) with a median DOR of 17.0 months (11.3 not estimated (NE)), and in the ALK-TKI pretreated patient population (*n* = 163), ORR was 56.4% (95% CI: 48.5–64.2), with a median DOR of 8.3 months (6.8–9.7). In the ALK-TKI naive cohort, mPFS was 18.4 months (11.1–NE), and in the patients who had previously received crizotinib, mPFS was 6.9 months (5.6–8.7). The phase II ASCEND-2 study evaluated the efficacy of 750 mg/day of ceritinib in 140 patients who had received two or more prior treatment regimens, including crizotinib. ORR was reported to be 38.6% (95% CI: 30.5–47.2%), median DOR was 9.7 months (95% CI: 7.1–11.1 months), and mPFS was 5.7 months (95% CI: 5.4–7.6 months). In the phase III ASCEND-4 study comparing 750 mg/day of ceritinib (*n* = 189) with platinum-based chemotherapy (*n* = 187) as first-line treatment, the primary endpoint, mPFS, was reported to be 16.6 months (95% CI: 12.6–27.2) in the ceritinib arm and 8.1 months (95% CI: 5.8–11.1) in the chemotherapy arm (HR: 0.55, 95% CI: 0.42–0.73) [28]. Median OS was reported to be NR (95% CI: 29.3–NE) in the ceritinib arm and 26.2 months (95% CI: 22.8–NE) in the chemotherapy arm (HR: 0.73 (95% CI: 0.50–1.08), *p* = 0.056). ORR was reported to be 72.5% (95% CI: 65.5–78.7) in the ceritinib arm and 26.7% (95% CI: 20.5–33.7) in the chemotherapy arm. Based on the results of the ASCEND-4 study, the FDA approved ceritinib as a first-line treatment for ALK-rearranged advanced NSCLC in 2017. The ASCEND-5 study compared 750 mg/day of ceritinib (*n* = 115) with chemotherapy (*n* = 116) (pemetrexed or docetaxel) in patients previously treated with chemotherapy and crizotinib. The primary endpoint, mPFS, was 5.4 months (95% CI: 4.1–6.9) vs. 1.6 months (95% CI: 1.4–2.8) in the ceritinib arm and the chemotherapy arm, respectively (HR: 0.49 (95% CI: 0.36–0.67)). ORR was 39.1% (95% CI: 30.2–48.7) in the ceritinib arm and 6.9% (95% CI: 3.0–13.1]) in the chemotherapy arm. Median OS was 18.1 months (95% CI: 13.4–23.9) in the ceritinib arm and 20.1 months (95% CI: 11.9–25.1) in the chemotherapy arm (HR: 1.0 (95% CI: 0.67–1.49), *p* = 0.50). In the phase II ASCEND-9 study, patients receiving ceritinib 750 mg/day with prior treatment with alectinib (*n* = 20) had an ORR of 25% (95% CI: 8.7–49.1) and mPFS of 3.7 months (95% CI: 1.9–5.3) [29]. Although there are no randomized clinical trials directly comparing ceritinib and crizotinib, an indirect analysis reported ceritinib prolongs mPFS (25.2 months vs. 10.8 months) (HR: 0.64 (95% CI: 0.47–0.87)). However, OS did not differ significantly, with HR of 0.82 (95% CI: 0.54–1.27) for ceritinib compared to crizotinib [30].

#### 3.2.1. Intracranial Efficacy

In the ASCEND-4 study, intracranial ORR in patients with measurable brain metastases at baseline was 72.7% (CR: 2, PR: 14) in the ceritinib arm (*n* = 22) and 27.3% (CR: 2, PR: 4) in the chemotherapy arm (*n* = 22). The median intracranial duration of intracranial response (IC-DOR) was 16.6 months (95% CI: 8.1–NE) in the ceritinib arm. The median IC-DOR could not be estimated in the chemotherapy arm because four of six patients had not progressed at analysis.

#### 3.2.2. Safety

The characteristic adverse events of ceritinib are gastrointestinal and hepatic disorders. In the ASCEND4 study (750 mg/day of ceritinib), diarrhea occurred in 85% (grades 3–4: 5%), nausea in 69% (grades 3–4: 3%), vomiting in 66% (grades 3–4: 5%), and increased ALT in 60% (grades 3–4: 31%). In the phase I ASCEND-8 study, a total of 306 patients were included and randomized to receive 450 mg with food (*n* = 108), 600 mg with food (*n* = 87), or 750 mg fasted (*n* = 111) of ceritinib. Of these, 304 patients were included in the safety analysis, and 198 untreated patients were included in the efficacy analysis. ORR was 78.1% (95% CI: 66.9–86.9), 72.5% (95% CI: 58.3–84.1), and 75.7% (95% CI: 64.3–84.9) for each dose groups. Median DOR was NE (95% CI: 11.2–NE), 20.7 (95% CI: 15.8–NE), and 15.4 (95% CI: 8.3–NE), respectively. In the safety analysis, the 450 mg dose group reported the lowest percentage of patients who had dose reductions (24.1%:65.1%:60.9%) and the lowest percentage of patients who exhibited gastrointestinal toxicity (75.9%:82.6%:91.8%). This result suggests that ceritinib 450 mg with food was shown to be non-inferior in efficacy to ceritinib 750 mg in fasted and better tolerated for gastrointestinal adverse effects [31]. In fact, in the ceritinib 750 mg/day group, diarrhea occurred in 64.4%, nausea in 62.2%, vomiting in 42.2%, and abdominal pain in 31.1%, while in the ceritinib 450 mg/day group, diarrhea occurred in 47.7%, nausea in 45.5%, vomiting in 22.7%, and abdominal pain in 22.7%.

Despite these efforts to counteract adverse events, ceritinib is still used less often than other ALK-TKIs due to gastrointestinal toxicity issues and the fact that crizotinib is not the control arm in the ASCEND-4 study, which led to its approval for first-line therapy.

### 3.3. Alectinib

Alectinib is a second-generation ALK-TKI and is currently one of the most commonly used first-line therapies. The initial phase III study was the J-ALEX study conducted in Japan [32,33]. A total of 207 Japanese patients in the J-ALEX study received either alectinib 300 mg twice daily (*n* = 103) or crizotinib 250 mg twice daily (*n* = 104) as first-line treatment. Median PFS, the primary endpoint, was 34.1 months (95% CI: 22.1-NE) in the alectinib arm and 10.2 months (95% CI: 8.3–12.0) in the crizotinib arm (HR: 0.37 (95% CI: 0.26–0.52)). In the final report of J-ALEX, there was no prolongation of mOS in the alectinib arm compared with the crizotinib arm (NR vs. 43.7 months, HR: 0.80 (95% CI: 0.35–1.82), *p* = 0.3860). The phase III ALEX trial, conducted with 303 treatment-naive, ALK-rearranged advanced NSCLC patients, compared alectinib at 600 mg twice daily (*n* = 152) with crizotinib at 250 mg twice daily (*n* = 151) in first-line treatment [34,35,36,37]. ORR was 82.9% (95% CI: 75.95–88.51) in the alectinib arm and 75.5% (95% CI: 67.84–82.12) in the crizotinib arm. In a recent report, the primary endpoint of mPFS was 34.8 months (95% CI: 17.7–NE) in the alectinib arm and 10.9 months (95% CI: 9.1–12.9) in the crizotinib arm (HR: 0.43 (95% CI: 0.32–0.58)). Immature OS data showed that mOS was NR with alectinib versus 57.4 months with crizotinib (HR: 0.67 (95% CI: 0.46–0.98)), indicating a statistically significant OS benefit of alectinib over crizotinib, despite 53.5% of patients receiving other ALK-TKIs by crossover after resistance to alectinib. The ALEX trial is still ongoing (NCT:02075840). In the phase III ALESIA, enrolling only Asian patients with ALK-rearranged advanced NSCLC, alectinib at 600 mg twice daily (*n* = 125) was compared with crizotinib at 250 mg twice daily (*n* = 62) as first-line treatment. The median PFS was NE vs. 11.1 months (HR: 0.22 (95% CI: 0.13–0.38)). ORR was 91% vs. 77%. DOR was longer in the alectinib arm than in the crizotinib arm (HR: 0.22 (95% CI: 0.12–0.40), *p* < 0.0001).

#### 3.3.1. Intracranial Efficacy

In the ALEX trial, alectinib was reported to have a better intracranial response compared to crizotinib. The CNS ORR in patients with measurable CNS metastases at baseline was 85.7% in the alectinib arm versus 71.4% in the crizotinib arm in patients with previous radiotherapy and 78.6% in alectinib arm versus 40.0% in crizotinib arm in patients without previous radiotherapy. In patients with measurable/not measurable baseline CNS metastases, CNS DOR was NR (95% CI: 14.8–NR) in the alectinib arm and 11.1 months (95% CI: 13.7–18.1) in the crizotinib arm in patients with prior radiotherapy. The CNS DOR in patients without previous radiotherapy was NR (95% CI: 13.4–NR) in the alectinib arm and 3.7 months (95% CI: 2.3–5.5) in the crizotinib arm, which was longer than that for crizotinib.

#### 3.3.2. Safety

The most common adverse events of all grades in the ALEX trial with a difference in frequency of 5% or more compared to crizotinib were anemia (20%), peripheral edema (17%), myalgia (16%), increased ALT (15%), increased AST (14%), increased blood bilirubin (15%), nausea (14%), and diarrhea (12%). The most common grade 3–5 adverse events were anemia (5%), increased ALT (5%), and increased AST (5%). In a recent report, both alectinib and crizotinib grade 3–5 adverse events (52.0% vs. 56.3%), adverse events leading to dose reduction (20.4% vs. 19.9%), adverse events requiring dose interruption (26.3% vs. 26.5%), and adverse events requiring treatment discontinuation (14.5% vs. 14.6%) were similar in frequency.

### 3.4. Lorlatinib

Lorlatinib, a third-generation ALK-TKI, was developed to target mutations conferring resistance to crizotinib and next-generation TKIs, and as a selective, brain-penetrating ALK-TKI. Unlike first- and second-generation ALK-TKIs, which are acyclic ALK-TKIs, lorlatinib is a macrocyclic ALK-TKI, which is smaller and more compact, making it sensitive to ALK mutations resistant to first- and second-generation ALK-TKIs, such as G1202R. In the phase II study, the primary endpoints were ORR and intracranial tumor response [38]. Patients with ALK-rearranged NSCLC were enrolled in the following expansions. No prior treatment (EXP1: *n* = 30), prior crizotinib treatment without chemotherapy (EXP2: *n* = 27), prior crizotinib treatment with chemotherapy (EXP3A: *n* = 32), prior non-crizotinib ALK-TKI with/without chemotherapy (EXP3B: *n* = 28), prior two ALK-TKI treatments with/without chemotherapy (EXP4: *n* = 66) or prior three ALK-TKI treatments with/without chemotherapy (EXP5: *n* = 46). In EXP1, ORR was 90.0% (95% CI: 73.5–97.9) and mPFS was NR (95% CI: 11.4–NR). In EXP2–5, ORR was 47.0% (95% CI: 39.9–54.2) and mPFS was 7.3 months (95% CI: 5.6–11.0). In EXP2–3A, ORR was 69.5% (95% CI: 56.1–80.8) and mPFS was NR (95% CI: 12.5–NR). In EXP3B, ORR was 32.1% (95% CI: 15.9–52.4) and mPFS was 5.5 months (95% CI: 2.7–9.0). In EXP4–5, ORR was 38.7% (95% CI: 29.6–48.5) and mPFS was 6.9 months (95% CI: 5.4–9.5). Based on the results of this trial, the FDA approved lorlatinib for patients with ALK-rearranged advanced NSCLC progressing on crizotinib and at least one other ALK-TKI, or after first-line treatment with non-crizotinib ALK-TKI, in November 2018. In the phase III CROWN trial, 296 untreated patients with ALK-rearranged advanced NSCLC were randomized to receive lorlatinib at 100 mg once daily (*n* = 149) or crizotinib at 250 mg twice daily (*n* = 147) [39]. The primary endpoint, mPFS, was NR vs. 9.3 months (95% CI: 7.6–11.1) at a median follow-up of 18.3 months for lorlatinib and 14.8 months for crizotinib (HR: 0.28 (95% CI: 0.191–0.413)). The proportion of patients who were alive without disease progression at 12 months was 78% (95% CI: 70–84) in the lorlatinib arm and 39% (95% CI: 30–48) in the crizotinib arm (HR: 0.28 (95% CI: 0.19–0.41), *p* < 0.001). ORR was 76% (95% CI: 68–83) for lorlatinib and 58% (95% CI: 49–66) for crizotinib. The HR for death was 0.72 (95% CI: 0.41–1.25) in the immature OS analysis at the data-cutoff point, which was not significant. Based on the result of the CROWN trial, the FDA approved lorlatinib as a first-line treatment for patients with ALK-rearranged NSCLC in March 2021. The CROWN trial is still ongoing (NCT:03052608).

#### 3.4.1. Intracranial Efficacy

Lorlatinib was developed for its potential effect on the CNS and has been reported to have excellent efficacy. Phase II results showed that intracranial ORR in EXP1 (*n* = 3) was 66.7% (95% CI: 9.4–99.2), with a median DOR of NR, and intracranial ORR in EXP2–5 (*n* = 198) was 63% (95% CI: 51.5–73.4), with a median DOR of 14.5 months (95% CI: 8.4–14.5). In the CROWN trial, intracranial ORR was 66% (95% CI: 49–80) in the lorlatinib arm (*n* = 38) and 20% (95% CI: 9–36) in the crizotinib arm (*n* = 40) among patients with measurable or non-measurable CNS metastases at baseline. Median DOR was NR in the lorlatinib arm and 9.4 months (95% CI: 6.0–11.1) in the crizotinib arm. The proportion of patients with intracranial response lasting longer than 12 months was 72% in the lorlatinib arm and 0% in the crizotinib arm. Of patients with measurable CNS metastases at baseline, intracranial ORR was 82% (95% CI: 57–96) in the lorlatinib arm (*n* = 17), 23% (95% CI: 5–54) in the crizotinib arm (*n* = 13), and CR was 71% and 8%, respectively. Median DOR was reported to be NE in the lorlatinib arm and 10.2 months (95% CI: 9.4–11.1) in the crizotinib arm.

#### 3.4.2. Safety

In the CROWN trial, adverse events that occurred more frequently (≥10%) in the lorlatinib arm than in the crizotinib arm were hypercholesterolemia (70%), hypertriglyceridemia (64%), edema (55%), weight gain (38%), peripheral neuropathy (34%), cognitive effects (21%), and diarrhea (21%). Hypercholesterolemia and hypertriglyceridemia can usually be easily managed with lipid-lowering drugs and dose modification. Cognitive effects and mood effects (16%) were typically grade 1 and reversible with dose interruption. Grade 3–4 adverse events occurred in 72% of patients treated with lorlatinib. The most common grade 3–4 adverse events in the lorlatinib arm were elevated triglycerides (20%), weight gain (17%), elevated cholesterol (16%), and hypertension (10%). Adverse events that led to withdrawal occurred in 49% of patients, dose reduction in 21%, and treatment discontinuation in 7%.

Lorlatinib has been an ALK-TKI frequently used after first-line treatment but based on the results of the phase III CROWN trial, lorlatinib is expected to be used more frequently in first-line treatment. Whether it is better to use lorlatinib after second-generation ALK-TKI treatment or in the first-line setting requires further study.

### 3.5. Brigatinib

In the phase II ALTA trial, the efficacy of brigatinib was evaluated in 222 patients with ALK-rearranged advanced NSCLC previously treated with crizotinib. Patients were randomized 1:1 to receive brigatinib 90 mg once daily (*n* = 112) or brigatinib 180 mg once daily with a 7-day lead-in at 90 mg (*n* = 110) [40]. The primary endpoint, ORR, was 46% (95% CI: 35–57%) and 56% (95% CI: 45–67%), respectively. Median DOR was 12.0 months (95% CI: 9.2–17.7) and 13.8 months (95% CI: 10.2–19.3), and mPFS was 9.2 months (95% CI: 7.4–12.8) and 16.7 months (95% CI: 11.6–21.4), for each dose. Median OS was 29.5 months (95% CI: 18.2–NR) and 34.1 months (95% CI: 27.7–NR), respectively. Following the results of the ALTA trial, in April 2017, the FDA approved brigatinib for the treatment of patients with ALK-rearranged NSCLC who have progressed on or are intolerant to crizotinib. The phase III ALTA-1L trial compared brigatinib (*n* = 137) with crizotinib (*n* = 138) as a first-line treatment [41,42,43]. In the interim analysis, ORR was 74% (95% CI: 66–81) vs. 62% (95% CI: 53–70), and DOR was NR (95% CI: 19.4–NR) vs. 13.8 months (95% CI: 9.3–20.8), respectively. The primary endpoint, mPFS, was 24.0 months (95% CI: 18.4–43.2) in the brigatinib arm and 11.1 months (95% CI: 9.1–13.0) in the crizotinib arm (HR: 0.48 (95% CI: 0.35–0.66)). Immature OS data at final analysis indicated similar OS in the two groups (HR: 0.81 (95% CI: 0.53–1.22), *p* = 0.305). However, OS may be affected by imbalances in subsequent anticancer therapy, such as a higher rate of crossover to brigatinib and a higher rate of subsequent anticancer therapy after discontinuation of study treatment in the crizotinib arm. A sensitivity analysis of OS adjusted for possible confounding by crossover suggested that treatment with brigatinib was associated with improved OS in the absence of treatment crossover in the crizotinib arm (HR = 0.54). Based on the results of the ALTA-1L trial, the FDA approved brigatinib as a first-line treatment for patients with ALK-rearranged advanced NSCLC in May 2020. The efficacy of brigatinib after second-generation ALK-TKI therapy has been evaluated in subsequent trials. In the subgroup analysis of the international expanded access program (EAP), the median time to discontinuation of brigatinib was 8.72 months (95% CI: 7.50–14.93) after treatment with alectinib (*n* = 111), 10.33 months (95% CI: 8.13–13.93) after treatment with ceritinib (*n* = 249), and 7.5 months (95% CI: 4.47–NE) after treatment with lorlatinib (*n* = 37) [44]. The phase II J-ALTA trial evaluated the efficacy and safety of brigatinib in 72 Japanese patients previously treated with ALK-TKIs, including alectinib. In 47 patients previously treated with alectinib, ORR was 34% (95% CI: 21–49), median DOR was 11.8 months (95% CI: 5.5–16.4), and mPFS was 7.3 months (95% CI: 3.7–9.3) [45]. The phase II ALTA-2 trial [46] (NCT:03535740) evaluating its efficacy as a subsequent treatment after treatment with alectinib or ceritinib and the phase III ALTA-3 trial [47] (NCT:03596866), comparing its efficacy with alectinib as a subsequent treatment after crizotinib are ongoing.

#### 3.5.1. Intracranial Efficacy

The interim analysis of the phase III ALTA-1L trial showed that in patients with measurable brain metastases, intracranial ORR was 78% (14/18) (95% CI: 52–94) for brigatinib and 26% (6/23) (95% CI: 10–48) for crizotinib. The final results showed that median intracranial DOR for patients with measurable brain metastases at baseline was 27.9 months (95% CI: 5.7–NE) in the brigatinib arm and 9.2 months (95% CI: 3.9–NE) in the crizotinib arm. In patients with brain metastases at baseline, the 3-year intracranial PFS rate was 31% (95% CI: 17–47) with brigatinib and 9% (95% CI: 2–25) with crizotinib (HR: 0.29, 95% CI: 0.17–0.51), and the 4-year rate was 22% (95% CI: 9–39%) with brigatinib and NE with crizotinib.

#### 3.5.2. Safety

Frequent adverse events reported in the ALTA-1L trial compared with crizotinib were diarrhea (52%), increased blood creatine phosphokinase (46%), cough (35%), hypertension (32%), nausea (30%), increased AST (26%), increased lipase (23%), increased ALT (21%), vomiting (21%), and dyspnea (21%). Common grade 3 or higher adverse events were increased blood creatine phosphokinase (24%), increased lipase (14%), and hypertension (12%). In the final analysis, 13% of adverse events led to treatment discontinuation, 44% led to dose reduction, and 72% led to dose interruption. Interstitial lung disease (ILD)/pneumonia is a characteristic adverse event of brigatinib. In the ALTA-1L trial, ILD/pneumonia occurred in 5% of patients (grade 3–4 ILD/pneumonia in 3%/1%), and ILD/pneumonia was reported to occur earlier after initiation of treatment in the ALTA study (median: day 2; range: days 1–9).

### 3.6. Ensartinib

In the phase I/II trial, the efficacy and safety of ensartinib were evaluated [48]. In dose escalation, patients with advanced solid tumors were administered ensartinib at doses of 25 to 250 mg once daily, and in dose expansion, patients with advanced ALK-rearranged NSCLC were administered additional doses of 225 mg once daily. ORR was 60% (95% CI: 47.4–71.4) and mPFS was 9.2 months (95% CI: 5.6–11.7) in 60 patients with ALK-rearranged NSCLC. In ALK-TKI-naive patients, ORR was 80%, and mPFS was 26.2 months (95% CI: 9.2-NE). In patients previously treated with crizotinib alone, ORR was 69%, and mPFS was 9.0 months (95% CI: 5.6–11.7). In the group previously treated with crizotinib and a second-generation ALK-TKI, mPFS was 1.9 months (95% CI: 1.7–5.7). The phase III eXalt3 study compared 225 mg of ensartinib once daily (*n* = 143) with 250 mg of crizotinib twice daily (*n* = 147) in 290 patients with ALK-rearranged untreated advanced NSCLC [49]. The primary endpoint, mPFS, was 25.8 months (95% CI: 0.03–44.0) in the ensartinib arm and 12.7 months (95% CI: 0.03–38.6) in the crizotinib arm (HR: 0.51 (95% CI: 0.36–0.72)). Median DOR was NR (95% CI: 22.0–NR) in the ensartinib arm and 27.3 months (95% CI: 12.9–NR) in the crizotinib arm. The eXalt3 study is currently ongoing (NCT:02767804). Ensartinib is expected to be approved by the FDA to treat ALK-rearranged advanced NSCLC based on the eXalt3 study.

#### 3.6.1. Intracranial Efficacy

In the phase I/II trial, 14 patients had CNS-targeted lesions at baseline. ORR of 64.3% (95% CI: 38.8–83.7) was reported with two patients with CR and seven patients with PR. In the eXalt3 study, intracranial ORR in patients with measurable brain metastases was 54% for ensartinib and 19% for crizotinib. In patients with intracranial disease, ORR was 64% for ensartinib and 21% for crizotinib, delaying the incidence of new central lesions in patients without baseline brain metastases (23.9 months vs. 4.2 months, HR: 0.32 (95% CI: 0.15–0.64)).

#### 3.6.2. Safety

In the interim analysis of the phase III eXalt3 study, rash (67.8%), increased AST (37.8%), increased ALT (48.3%), pruritus (26.6%), nausea (22.4%), constipation (20.3%), edema (21.0%), anemia (14.0%), vomiting (11.9%), increased ALP (13.3%), increased blood creatinine (14.0%), increased γ-GTP (13.3%), and anorexia (11.2%) were reported. Grade 3 rash was reported in 11.2% of patients, requiring dose interruption or reduction. Dose reduction in 24% of patients and discontinuation of treatment in 9% were required due to adverse events, such as rash and liver toxicity.

The summary of six TKIs is in the tables below (Table 1, Table 2, Table 3 and Table 4).

## 4. Combination Therapy with Angiogenesis Inhibitors and ALK-TKIs

Angiogenesis inhibitors prevent tumor growth by blocking the signals that promote tumor angiogenesis. There are no clinical trials evaluating the efficacy of angiogenesis inhibitors monotherapy in ALK-rearranged NSCLC. The efficacy of ramucirumab plus erlotinib in EGFR-mutated advanced NSCLC has been reported [50]. Similarly, combination therapy with angiogenesis inhibitors and ALK-TKIs has been studied. A single-arm, prospective observational study investigating the efficacy and safety of bevacizumab plus crizotinib in ALK/ROS-1/c-MET-positive advanced NSCLC has been reported [51]. Patients received crizotinib (250 mg twice daily) and bevacizumab (7.5 mg/kg every 3 weeks) until disease progression or intolerable toxicity or death. The primary endpoint was mPFS, and the secondary endpoints were DOR, ORR, disease control rate (DCR), and safety. In 12 patients with ALK-rearranged NSCLC, mPFS was 13.9 months, and median DOR was 14.8 months. Median OS was NR, and the 3-year survival rate was 79.5%. Of the 12 patients, the best overall response was PR in seven and SD in five. ORR and DCR were 58.3% and 100%, respectively. The most common adverse events were fatigue (28.6%) and rash (21.4%). Other adverse events reported were nausea (14.3%), vomiting (7.1%), edema (7.1%), and pain (7.1%). Most adverse events (86.7%) were grade 1–2, but two patients had grade 3 or 4 elevations in aminotransferases, and both discontinued treatment. In addition, one patient reported grade 1 hemoptysis, and treatment was discontinued. No interstitial lung disease, active bleeding, hypertension, or treatment-related death occurred. Though this study initially planned to enroll 30 or more patients, however, during the enrollment, second-generation ALK inhibitors were approved as first-line treatment, which made the enrollment difficult, and the sample size was small. However, several other combination therapies with angiogenesis inhibitors and ALK-TKIs are also ongoing (NCT:03779191, NCT:02521051: bevacizumab plus alectinib), (NCT:04227028: bevacizumab plus brigatinib), (NCT:04837716: carboplatin, pemetrexed, bevacizumab plus ensartinib).

## 5. Immune Checkpoint Inhibitor

An immune checkpoint inhibitor (IO) that targets programmed cell death-1 (PD-1), programmed cell death ligand-1 (PD-L1), and CTLA-4 is an important therapeutic agent for NSCLC. The presence of EML4-ALK is associated with increased PD-L1 expression by activating the PI3K-AKT and MEK-ERK pathways [52]. Therefore, treatment of ALK-rearranged NSCLC with immune checkpoint inhibitors is expected to be an effective therapeutic strategy. However, IO has been reported to lack efficacy in NSCLC with oncogenic driver mutations, such as EGFR and ALK. In a single-center retrospective study evaluating NSCLC patients with EGFR-mutant (*n* = 22) or ALK-rearranged (*n* = 6) NSCLC treated with IO, objective response was observed in only one patient with EGFR mutation and ORR in patients with ALK-rearrangement was 0%, in contrast to 7 of 30 patients (23.3%) with EGFR wild type or ALK-negative/unknown (*p* = 0.053) [53]. Another multicenter retrospective study reported an ORR of 0%, mPFS of 2.5 months (95% CI: 1.5–3.7), and mOS of 17.0 months (95% CI: 3.6–NR) for 23 NSCLC patients with ALK-rearrangement treated with IO monotherapy [54]. In a retrospective real-world study of 83 patients (of these, 74 patients received IO as monotherapy) with ALK-rearranged NSCLC who received IO from a multicenter electronic medical record-derived database, the mPFS of patients who received IO before ALK-TKI was 3.9 months, and that of patients who received IO after ALK-TKI was 1.5 months [55]. CheckMate 057 [56] and KEYNOTE-010 [57] are prospective clinical trials treating NSCLC patients with IO, including ALK-rearranged NSCLC patients. However, due to the small number of patients in both trials, outcomes for the ALK-positive cohort have not been reported. Combination therapy with ALK-TKIs and IO has been evaluated in several trials, but many trials have resulted in poor efficacy and increased toxicity. Group E of the phase I/II CheckMate 370 evaluated the safety of the combination of nivolumab (240 mg once every two weeks) and crizotinib (250 mg twice daily) as first-line treatment for previously untreated ALK-rearranged advanced NSCLC patients. Of the 13 patients, 5 (38%) developed severe hepatotoxicity and discontinued the combination therapy, and 2 of the 5 patients died. ORR was reported to be 38% [58]. In the phase Ib study [59] of pembrolizumab plus crizotinib for previously untreated ALK-rearranged advanced NSCLC, one of the first two patients enrolled at dose level 0 (crizotinib 250 mg twice daily and pembrolizumab 200 mg every 3 weeks) required discontinuation of pembrolizumab due to grade 3 liver toxicity, and one died of grade 4 pneumonia induced by pembrolizumab. At dose level -1 (3 weeks of crizotinib monotherapy 250 mg twice daily, followed by crizotinib 250 mg twice daily, with the addition of pembrolizumab 200 mg every 3 weeks), two of the seven patients required pembrolizumab discontinuation and crizotinib discontinuation/reduction due to liver toxicity. This trial was terminated early in the dose-finding phase due to the severe adverse events identified. The phase Ib study evaluated nivolumab (3 mg/kg every 2 weeks) in combination with ceritinib (*n* = 14: 450 mg/day or *n* = 22: 300 mg/day) in 36 patients with previously treated or untreated stage IIIB/IV ALK-rearranged NSCLC [60]. In the dose-escalation study, ORR for patients without prior treatment with ALK-TKI was 83% (95% CI: 35.9–99.6) in the 450 mg/day ceritinib group and 60% (95% CI: 26.2–87.8) in the 300 mg/day ceritinib group, and ORR for patients with previous treatment with ALK-TKI was 50% (95% CI: 15.7–84.3) in the 450 mg/day group and 25% (95% CI: 5.5–57.2) in the 300 mg/day group. Elevated alanine aminotransferase level (25%), elevated gamma-glutamyl transferase level (22%), elevated amylase level (14%), elevated lipase level (11%), and maculopapular rash (11%) were reported to be common grade 3–4 adverse events. The protocol had to be changed for toxicity management, and eventually, the registration was terminated. A phase I trial of erlotinib or crizotinib in combination with the CTLA-4 inhibitor ipilimumab was conducted in patients with EGFR-mutant or ALK-rearranged NSCLC [61]. Median PFS for the three patients with ALK-rearranged NSCLC was 24.1 months. One of the three patients developed dropsy, and one developed grade 2 pneumonia, and the trial was terminated early. Trials of alectinib plus atezolizumab (NCT:02013219) have been completed, and results are awaited. Two clinical trials of combination therapy of platinum-doublet with IO (+/− angiogenesis inhibitor) for EGFR-mutant/ALK-rearranged NSCLC are ongoing (NCT:04042558: platinum-pemetrexed-atezolizumab +/− bevacizumab), along with the trial of (NCT:03991403: atezolizumab + bevacizumab + carboplatin + paclitaxel). The results of IO monotherapy and the combination of IO + TKI are summarized in Table 5 and Table 6, respectively.

## 6. Chemotherapy

In a previously reported study, pemetrexed has been reported to be effective for ALK-rearranged NSCLC. In the PROFILE 1007 study, median PFS in the chemotherapy arm was 4.2 months in the pemetrexed arm and 2.6 months in the docetaxel arm, suggesting that pemetrexed may be more effective than docetaxel. In addition, two retrospective studies showed that pemetrexed prolongs PFS in ALK-rearranged NSCLC. In a retrospective study of 89 advanced-NSCLC patients (ALK-positive: 19, EGFR-mutant: 12, KRAS-mutant: 21, and wild type: 37), mPFS was reported as 9 months (95% CI: 3–12) in the ALK-positive, 5.5 months (95% CI: 1–9) in the EGFR mutant, 7 months (95% CI: 1.5–10) in the KRAS mutant, and 4 months (95% CI: 3–5) in the wild type. In the multivariate analysis in this study, ALK was the only driver mutation associated with prolonged PFS in the chemotherapy regimen, including pemetrexed (HR: 0.36 (95% CI: 0.17–0.73), *p* = 0.0051) [62]. A retrospective study of 95 patients with advanced NSCLC (ALK-positive: 43, EGFR-mutant: 15, wild-type: 37) reported the efficacy of pemetrexed [63]. ORR was 46.7% in ALK-positive, 16.2% in the EGFR mutant, and 4.7% in the KRAS mutant. Time to progression was 9.2 months in the ALK-positive, 1.4 months in the EGFR mutant, and 2.9 months in the KRAS mutant, regardless of treatment line. ALK-positive was shown to be a significant predictor of ORR (HR: 0.07 (95% CI: 0.01–0.32), *p* = 0.001) and time to progression (HR: 0.44 (95% CI: 0.24–0.80), *p* = 0.007). Pemetrexed is an important therapeutic agent for ALK-rearranged NSCLC that is not responsive to ALK-TKIs or cannot tolerate adverse events. The efficacy of combination therapy with EGFR-TKI and platinum doublet therapy for EGFR-mutant NSCLC has been reported in two phase III trials [64,65]. Similarly, the combination of pemetrexed and ALK-TKI is well tolerated; it may improve PFS and OS. The phase II trial to evaluate the efficacy of combination therapy of platinum doublet and ALK-TKI in ALK-rearranged NSCLC is ongoing in Japan (jRCTs041210103). Recent and ongoing clinical trials are summarized in Table 7.

## 7. Drug Sensitivity

Many ALK fusion partners, such as EML4, PM-3/-4, CLTC, LMNA, PRKAR1A, RANBP2, TFG, FN1, and KIF5B, have been identified [66,67]. It is known that the activity of ALK-TKIs varies depending on the partner; for instance, KIF5B-ALK was highly sensitive to ensartinib but was one of the least sensitive fusions to crizotinib and lorlatinib [68]. In addition, several EML4-ALK variants have been reported [69]. The most frequent variant is variant 1, in which exon 13 of EML4 is fused to exon 20 of ALK (E13; A20), and the next most frequent variant is variant 3a/b, in which exon 6a or 6b of EML4 is fused to exon 20 of ALK (E6a/b; A20). Variant 3 is known to have a shorter mPFS than variant 1 with treatment with crizotinib, alectinib, and ceritinib [70,71]. The ALTA-1L trial evaluated the efficacy of each variant of brigatinib, and similarly, variant 3 showed poorer outcomes compared to variant 1 [43]. The poor prognosis of variant 3 has been attributed to the shorter variant being more stable, accumulating in greater numbers, and interacting better with the cytoskeleton, causing stronger oncogenic signaling, less sensitivity to ALK-TKIs, and accelerated migration and metastasis [72,73,74]. In a retrospective study (*n* = 129), resistance mutations were identified in 10 patients (30%) in variant 1 and 25 patients (57%) in variant 3 (*p* = 0.023). In particular, the G1202R mutation was detected in 32% (14/44) in variant 3 compared to 0% (0/33) in variant 1 (*p* < 0.001). In the same study, an analysis of 12 patients with variant 1 and 17 patients with variant 3, who received lorlatinib after treatment failure with both crizotinib and at least one second-generation ALK-TKI, reported that patients with variant 3 had a significantly longer PFS than patients with variant 1 (mPFS: 11.0 months vs. 3.3 months, HR: 0.31 (95% CI: 0.12–0.79), *p* = 0.011) [75]. The higher drug sensitivity of variant 3 to lorlatinib than variant 1 is because ALK mutations, including the G1202R mutation, which is effective for lorlatinib, are more likely to be associated with variant 3. The TP53 mutation is a common gene mutation responsible for ALK-TKI resistance [76]. A retrospective study showed that the presence of the TP53 mutation reduced sensitivity to ALK-TKIs, and patients with both variant 3 and the TP53 mutation had a poor prognosis [77,78]. The efficacy of the combined use of a proteasome inhibitor with alectinib in ALK-rearranged NSCLC cells with TP53 mutation has been shown in vitro and is expected to be applied to clinical practice [79].

## 8. Mechanism of Resistance against ALK-TKI

The prognosis of ALK-rearranged advanced NSCLC has improved with the contribution of various therapeutic agents; however, cancer cells develop resistance, and patients eventually progress. There are two types of resistance to ALK-TKI-targeted therapy: primary resistance and acquired resistance.

### 8.1. Primary Resistance Mechanisms

Primary resistance is a rare condition, but according to case reports, primary resistance in patients with ALK-rearranged NSCLC is caused by ALK mutation [80], MYC amplification [81], EGFR mutation [82], a low mutant allele fraction (MAF) of the EML4-ALK-rearrangement [83], K-RAS mutations [84], and Bim deletion polymorphism [85].

### 8.2. Acquired Resistance Mechanisms

Mechanisms of acquired resistance are classified into ALK-dependent resistance and ALK-independent resistance.

#### 8.2.1. ALK-Dependent Resistance Mechanisms

ALK-dependent resistance mechanisms include ALK amplification and ALK mutation. ALK amplification has been reported as a resistance mechanism to crizotinib, but its incidence is low, and ALK mutation is the problem in most cases [76]. G1202R mutation is the most common ALK resistance mutation to second-generation ALK-TKIs, but the frequency of ALK resistance mutations depends on the ALK-TKI as prior therapy. A study reported the frequency of ALK mutations by performing biopsies after developing resistance to ALK-TKIs [76]. The common secondary ALK mutations were G1202R (21%), F1174 C/L (17%), and C1156Y (8%) after ceritinib; G1202R (29%), I1171T/S (12%), V11180L (6%), and L1196M (6%) after alectinib; and G1202R (43%), E1210K (29%), D1203N (14%), and S1206Y/C (14%) after brigatinib, respectively. Furthermore, this study presented in vitro IC50 values for ALK-TKIs regarding the different mutations. For instance, IC50 of alectinib for I1171T/S is >50/<200 nM, while the IC50 of ceritinib and brigatinib is reported to be <50. Therefore, subsequent treatment with other ALK-TKIs may be effective depending on ALK resistance mutations. The third-generation ALK-TKI, lorlatinib, has the broadest spectrum for single ALK resistance mutation, including the G1202R mutation. The efficacy of lorlatinib after treatment with a second-generation ALK-TKI was analyzed in 198 patients enrolled in the phase II study, and the prognosis was indicated to be different depending on whether the ALK mutation was the mechanism of resistance to previous therapy. ORR and PFS were 69% and 11.0 months, respectively, in the cohort in which ALK mutations were detected by tissue genotyping, while ORR and mPFS were 27% and 5.4 months in the ALK mutation-negative cohort [86]. The compound ALK mutation (for instance, G1269A + I1171S/C1156, G1202R + L1196M/F1174L, and L1196M + D1203N), which is the main cause of lorlatinib resistance, is the most clinically important major unmet need [87]. The fourth-generation ALK-TKIs are currently considered the most promising treatment for compound ALK mutation. Two fourth-generation ALK TKIs (TPX-0131 and NVL-655) are under development [88,89,90,91]. Both TPX-0131 and NVL-655 can inhibit acquired compound ALK mutations in addition to a wide spectrum of single ALK mutations. A phase I/II clinical trial of TPX-0131 for previously treated ALK-rearranged NSCLC patients (*n* = 210) is currently ongoing (NCT:04849273).

#### 8.2.2. ALK-Independent Resistance Mechanisms

ALK-independent resistance mechanisms include the activation of bypass signaling pathways, overexpression of *p*-glycoprotein (*p*-gp) [92], histological transformation [93], and epithelial–mesenchymal transition (EMT) [94]. The activation of bypass signaling pathways includes EGFR signaling [95,96], amplification of KIT [97], the IGF-1R-IRS-1 pathway [98], MAPK [99], MET amplification [100,101], BRAF V600E mutation [100], activation of the transcriptional co-regulator YAP [102], and HER2-amplification [103]. These can occur during treatment with any ALK-TKI, including lorlatinib, and can lead to resistance to ALK-TKIs. Combination therapy with ALK-TKI and their target drugs in patients with activation of bypass signaling pathways other than ALK may overcome drug resistance; thus, their detection is important [87,104,105]. MET amplification is sensitive to crizotinib. The efficacy of combination therapy with alectinib and crizotinib has been reported in patients in whom MET amplification was detected after progression with alectinib [106]. EMT is the morphological change in which cell-to-cell contacts are lost, making them more mobile and invasive, and tumor cells acquire a mesenchymal morphology and develop drug resistance. HDAC inhibitors have been reported to overcome this resistance mechanism by reversing EMT in vivo and in vitro [94].

### 8.3. Treatment Algorithm for ALK-Rearranged Advanced NSCLC

To date, five ALK-TKIs have been approved by the FDA. However, there are no definitive opinions or the results of clinical trials on sequencing; thus it is largely a matter of clinician judgment today. According to NCCN guideline 2022, alectinib, brigatinib, and lorlatinib are the preferable first-line ALK-TKIs. Ceritinib has not been directly compared with crizotinib in clinical trials, and gastrointestinal toxicity is an issue, although dosage and dose have been adjusted. Crizotinib has been shown to be inferior to other next-generation ALK-TKIs in systemic and central nervous system effects. When the disease progresses during first-line therapy, the identification of resistance mechanisms by performing tissue/liquid biopsy may help to guide optimal treatment. For example, in the case that EML4-ALK G1202R is the cause of resistance, lorlatinib may be a favorable subsequent therapy, and the fourth-generation ALK-TKIs to be developed in the future are effective for compound ALK mutation. When an EGFR mutation is identified, combination therapy with ALK-TKI and EGFR-TKI may be effective, and crizotinib is a reasonable option for patients with confirmed MET amplification. If the disease has converted to small cell lung cancer, a different chemotherapy regimen is required from those for NSCLC. Figure 2 shows the proposed treatment algorithm for ALK-rearranged advanced NSCLC.

## 9. Conclusions

We described the efficacy and safety of six ALK-TKIs, other chemotherapies, and resistance mechanisms. Currently, clinicians have four options as TKIs for first-line treatment: alectinib, ceritinib, lorlatiniba, and brigatinib. In addition, ensartinib is expected to be approved by the FDA in the future. However, there have been no clinical trials directly comparing second- and third-generation ALK-TKIs, of which TKI is the best, which is an issue that needs to be addressed. Furthermore, there are no definite conclusions regarding the treatment sequence after first-line therapy, and further investigation is required. Combination therapy of ALK-TKIs and angiogenesis inhibitors may become an important treatment regimen, as combination therapy of EGFR-TKIs and angiogenesis inhibitors has shown efficacy for a part of EGFR-mutant NSCLC patients. Furthermore, an important matter for angiogenesis inhibitors is that their efficacy may be enhanced when used in combination with IO. The results of clinical trials have shown that IO alone or in combination with IO and platinum doublet chemotherapy is poorly effective, and the combination of TKI and IO was too toxic to continue. The clinical trials of IO in combination with platinum doublet and angiogenesis inhibitors for advanced EGFR/ALK-positive NSCLC are ongoing, and the results are awaited. Resistance against ALK-TKIs is an important issue. Clinical trials of fourth-generation TKI capable of overcoming compound ALK mutation, which is the major mechanism of resistance to lorlatinib, are currently ongoing and are expected to eventually be approved for a treatment option. Moreover, the approach to the ALK-independent resistance mechanism is also important. Especially for patients with activation of bypass signaling pathways, the efficacy of anticancer agents targeting specific genetic mutations in combination with ALK-TKIs has been reported. However, activation of the bypass signaling pathway is rarely identified when the disease progresses in clinical practice, in part because tissue rebiopsy is not often performed. Liquid biopsy is one of the less-invasive methods to identify resistance mechanisms, and non-inferior results compared to tissue samples have been reported. We expect that advances in anticancer therapy and tools for identifying resistance mechanisms, including liquid biopsy, will improve the prognosis of ALK-rearranged NSCLC.

## Figures and Tables

**Figure 1 cancers-14-01184-f001:**
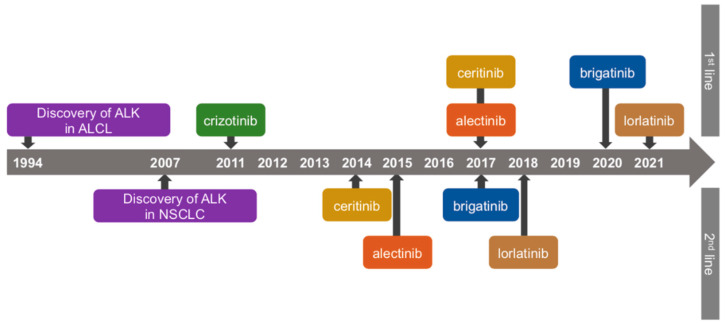
Timeline of discovery of ALK and US FDA approval of ALK-TKIs. ALCL: anaplastic large cell lymphoma, NSCLC: non-small cell lung cancer.

**Figure 2 cancers-14-01184-f002:**
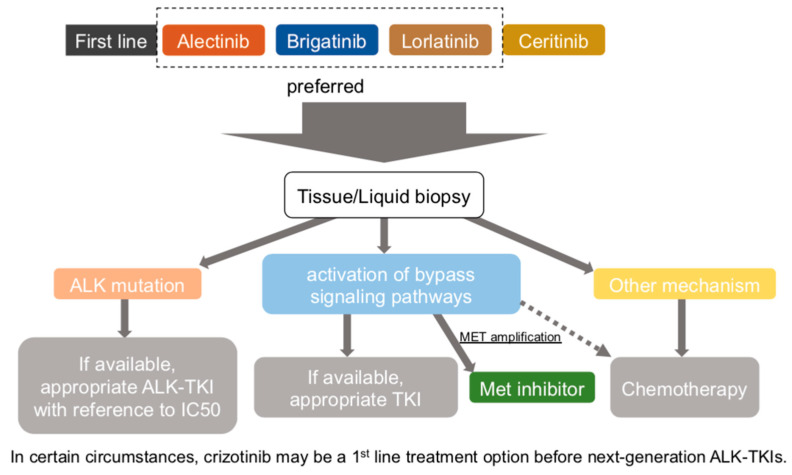
Proposed treatment algorithm for ALK-rearranged advanced NSCLC.

**Table 1 cancers-14-01184-t001:** Clinical trials as first-line treatment.

Clinical Trial	ALK-TKI	Control Arm	Patients (n)	ORR (%)	mPFS (Months)	PFS HR (95% CI)	Median Follow-Up Duration for PFS	mOS (Months)	OS HR (95% CI)	Median Follow-Up Duration for OS	Ref
PROFILE1014	Crizotinib	platinum doublet	172/171	74 vs. 45	10.9 vs. 7.0	0.45 (0.35–0.60)	NA	NR vs. 47.5	0.76 (0.55–1.05)	45.7 months	[22,23]
ASCEND-4	Ceritinib	platinum doublet	189/187	72.5 vs. 50	16.6 vs. 8.1	0.55 (0.42–0.73)	12.4 months	NR vs. 26.2	0.73 (0.50–1.08)	NA	[28]
ALEX	Alectinib	Crizotinib	152/151	82.9 vs. 75.5	34.8 vs. 10.9	0.43 (0.32–0.58)	37.8 months	NR vs. 57.4	0.67 (0.46–0.98)	48.2 months	[34,35,36,37]
J-ALEX	Alectinib	Crizotinib	103/104	92 vs. 79	34.1 vs. 10.2	0.37 (0.26–0.52)	42.4 months	NR vs. 43.7	0.80 (0.35–1.82)	NA	[32,33]
CROWN	Lorlatinib	Crizotinib	149/147	76 vs. 58	NR vs. 9.3	0.28 (0.19–0.41)	18.3 months	NR	0.72 (0.41–1.25)	NA	[39]
ALTA-1L	Brigatinib	Crizotinib	137/138	74 vs. 62	24.0 vs. 11.1	0.48 (0.35–0.66)	40.4 months	NR	0.81 (0.53–1.22)	NA	[41,42,43]
eXalt3	Ensartinib	Crizotinib	143/147	74 vs. 67	25.8 vs. 12.7	0.51 (0.36–0.72)	23.8 months	NR	NA	NA	[49]

ORR: overall response rate, mPFS: median progression-free survival, median overall survival.

**Table 2 cancers-14-01184-t002:** Efficacy following second-generation TKI treatment.

Clinical Trial	ALK-TKI	Treatment Line	Prior Treatment	Patients (n)	ORR (95% CI)	mPFS (Months) (95% CI)	Ref
ASCEND-9	Ceritinib	≥2	Alectinib (+Chemo or Crizotinib)	20	25% (8.7–49.1)	3.7 (1.9–5.3)	[29]
Phase II study	Lorlatinib	≥2	EXP3B *	28	32.1% (15.9–52.4)	5.5 (2.7–9.0)	[38]
			EXP4-5 *	111	38.7% (29.6–48.5)	6.9 (5.4–9.5)	
the international EAP	Brigatinib	≥2	**at least one ALK inhibitor**			**Time to treatment**	
						**discontinuation**	
			Alectinib	111	NA	8.72 (7.50–14.93)	[44]
			Ceritinib	249	NA	10.33 (8.13–13.62)	
			Lorlatinib	37	NA	7.5 (4.47–NE)	

* EXP3B: Previous non-crizotinib ALK-TKI with or without chemotherapy, * EXP4-5: ≥2 previous ALK-TKIs with or without chemotherapy.

**Table 3 cancers-14-01184-t003:** Efficacy for measurable brain metastases in first-line treatment.

Clinical Trial	ALK-TKI		Control Arm	Patients (n)	IC-ORR (%)	Median IC-DOR (Months) (95% CI)	Ref
ALEX	Alectinib	RT +	Crizotinib	7/7	85.7 vs. 71.4	NR (14.8–NR) vs. 11.1 (13.7–18.1)	[37]
		RT −	Crizotinib	14/15	78.6 vs. 40.0	NR (13.4–NR) vs. 3.7 (2.3–5.5)	
ASCEND-4	Ceritinib		platinum doublet	22/22	72.7 vs. 27.3	16.6 (8.1–NE) vs. NE	[28]
CROWN	Lorlatinib		Crizotinib	17/13	82 vs. 23	NE vs. 10.2 (9.4–11.1)	[39]
ALTA	Brigatinib		Crizotinib	18/23	78 vs. 26	27.9 (5.7–NE) vs. 9.2 (3.9–NE)	[40]
eXalt3	Ensartinib		Crizotinib	13/21	54 vs. 19	NA	[49]

RT: radiation therapy, IC-ORR: intracranial overall response rate, IC-DOR: intracranial duration of response.

**Table 4 cancers-14-01184-t004:** Characteristics of molecule and common/typical adverse effects.

ALK-TKI	Crizotinib			Ceritinib			Alectinib			Lorlatinib			Brigatinib			Ensartinib		
Molecular formula	C21H22Cl2FN5O			C28H36ClN5O3S			C30H34N4O2 HCl			C21H19FN6O2			C29H39ClN7O2P			C26H27Cl2FN6O3		
Characteristics of molecule	acyclic			acyclic			acyclic			macrocyclic			acyclic			NA		
Dosage	250 mg twice/day			450 mg once/day(fed)			600 mg twice/day			100 mg once/day			180 mg once/day (7 day lead-in at 90 mg/day)			225 mg once/day		
AEs (%)		any grade	grade ≧ 3		any grade	grade ≧ 3		any grade	grade ≧ 3		any grade	grade ≧ 3		any grade	grade ≧ 3		any grade	grade ≧ 3
	Vision disorder	73.1	0.6	Diarrhea	57.4	0.9	Conspitation	36.8	0.0	Hypercholesterolemia	70	16	Diarrhea	58	2	Rash	59.4	11.2
	Diarrhea	65.5	2.9	Vomiting	38.9	1.9	Anemia	26.3	5.9	Hypertriglyceridemia	64	20	Elevated CPK	50	26	Elevated ALT	46.2	4.2
	Nausea	59.1	1.8	Nausea	41.7	0	Fatigue	22.4	0	Edema	55	4	Nausea	33	2	Elevated AST	37.1	0.7
	Edema	52.6	1.2	Elevated AST	40.7	17.6	Elevated blood bilirubin	21.7	2.6	Increased weight	38	17	Hypertention	32	14	Constipation	31.5	0
	Vomiting	50.9	2.3	Elevated ALT	35.2	7.4	Peripheral edema	19.1	0	Peripheral neuropathy	34	2	Elevated AST	26	4	Cough	30.1	0.7
	Constipation	45.6	1.8	Elevated γ-GTP	33.3	22.2	Elevated ALT	17.8	4.6	Cognitive effects	21	2	Back pain	26	0	Prurtius	28.0	2.1
	Upper respiratory infection	39.8	0	Fatigue	22.2	0.9	Elevated AST	17.1	5.3	Hypertension	18	10	Elevated lipase	24	15	Nausea	26.6	1.4
	Elevated transaminases	39.2	14	Abdominal pain	20.4	0	Myalgia	17.1	0	Vision disorder	18	0	Elevated ALT	23	4	Edema	25.2	2.1
	Decreased appetite	35.1	2.3	Decreased appetite	18.5	0	Nausea	16.4	0.7	Mood effects	16	1	Pneumonia	10	5	Anemia	21.7	0.7

**Table 5 cancers-14-01184-t005:** Results of a retrospective study of IO monotherapy.

Study Design	Treatment Line	Patients (n)	ORR (%)	mPFS (Months)	mOS (Months)	Ref
retrospective	prior lines: median 3 (0–8)	6	0	NA	NA	[53]
retrospective	NA	23	0	2.5 (95% CI: 1.5–3.7)	17.0 (95% CI: 3.6–NR)	[54]
retrospective	1st:16 (19.2%), 2nd:25 (30.1%), ≥3rd:42 (50.6%)	83 (IO monotherapy:74)	NA	before TKI (*n* = 42):3.9 m after TKI (*n* = 41):1.5 m	NA	[55]

**Table 6 cancers-14-01184-t006:** Results of clinical trials of IO + TKI combination therapy.

Phases	IO	Treatment Line	Patients (n)	ORR (%)	mPFS (Months)	AE ≥ Grade3 (%)	Status	Ref
I/II	nivolumab + crizotinib	first	13	38	NA	62	Completed	[58]
Ib	pembrolizumab + crizotinib	first	dose level 0:2	50	NA	100	Terminated	[59]
			dose level −1:7	57	NA	29		
Ib	nivolumab + ceritinib (450 mg)	ALK-TKI naïve	6	83 (95% CI: 35.9–99.6)	NR (95% CI: 1.8–NE)	93	Active, not recruiting	[60]
		ALK-TKI pretreated	8	50 (95% CI: 15.7–84.3)	6.4 (95% CI: 0.8–13.7)			
	nivolumab + ceritinib (300 mg)	ALK-TKI naïve	10	60 (95% CI: 26.2–87.8)	NR (95% CI: 1.9–NE)	82		
		ALK-TKI pretreated	12	25 (95% CI: 5.5–57.2)	3.7 (95% CI: 1.8–NE)			

**Table 7 cancers-14-01184-t007:** Recent and ongoing clinical trials.

	Interventions	Phases	Enrollment	Status
**Combination therapy with angiogenesis inhibitors and ALK-TKI**				
NCT:03779191	Alectinib + Bevacizumab	2	40	Recruiting
NCT:02521051	Alectinib + Bevacizumab	1/2	43	Recruiting
NCT:04227028	Bevacizumab + Brigatinib	1	31	Recruiting
NCT:04837716	Bevacizumab + Carboplatin + Ensartinib + Pemetrexed	1	12	Recruiting
**Imune checkpoit inhibitor + ALK-TKI**				
NCT:02013219	Alectinib or Erlotinib + Atezolizumab	1	52	Completed
**Plutinum doubulet + bevacizumab + Imune checkpoit inhibitor**				
NCT:04042558	Carboplatin + Pemetrexed + Atezolizumab + Bevacizumab vs. Carboplatin + Pemetrexed + Atezolizumab	2	149	Recruiting
NCT:03991403	Atezolizumab + Bevacizumab + Carboplatin or Cisplatin + Pemetrexed or Paclitaxel	3	228	Recruiting
**Chemotherapy + ALK-TKI**				
jRCTs041210103	Carboplatin + Pemetrexed +Brigatinib	2	110	Recruiting
**ALK-TKI monotherapy**				
NCT:03535740	Brigatinib	2	103	Active, not recruiting
NCT:0359686	Brigatinib vs. Alectinib	3	246	Recruiting

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
