# Peer review of "Review of Therapeutic Strategies for Anaplastic Lymphoma Kinase-Rearranged Non-Small Cell Lung Cancer"

_cancers, 2022, doi:10.3390/cancers14051184_

Round 1

Reviewer 1 Report

The authors reviewed the recent clinical trials of ALK-TKIs, angiogenesis inhibitors, and immune checkpoint inhibitors and to describe tissue and liquid biopsy as a method to investigate the mechanisms of resistance against ALK-TKIs. There are several major problems:

  1. There are many ROS1 studies, but the authors barely mentioned them.
  2. The manuscript was poorly organized. The levels of tittles were not clear. For example, under “2.1 Crizotinib”, there were “Intracranial efficacy” and “Safety”. But “Intracranial efficacy” and “Safety” should be “2.1.1 Intracranial efficacy” and “2.1.2 Safety”.
  3. The authors only listed the statistics form previous studies and did not provide their own perspectives or opinions.
  4. There were no figures. The authors should summary the mechanisms as a figure and add a timeline figure of key findings according years.
  5. The authors describe each drug. What was the mechanisms of these drugs? Can they be categorized?

Author Response

Response to Reviewer 1 Comments
We thank the reviewer for insightful comments, which we feel have helped us to improve our manuscript. Our specific responses to the points raised are as follows:

Point 1: There are many ROS1 studies, but the authors barely mentioned them.

Response 1:  This manuscript is intended only for ALK-positive lung cancer. Therefore, studies of ROS-1 are not included. However, the results of the PROFILE 1001 study in ROS1-positive patients are important in the context of the development of crizotinib. We added description of crizotinib as a multi-kinase inhibitor, including ROS1, and the PROFILE 1001 study to Page3 line 149-154.

Point 2: The manuscript was poorly organized. The levels of tittles were not clear. For example, under “2.1 Crizotinib”, there were “Intracranial efficacy” and “Safety”. But “Intracranial efficacy” and “Safety” should be “2.1.1 Intracranial efficacy” and “2.1.2 Safety”.

Response 2: We fully agree with reviewer’s suggestion. However, "MDPI editor" has already edited it for us.

Point 3: The authors only listed the statistics form previous studies and did not provide their own perspectives or opinions.

Response 3: We agree with reviewer’s suggestion and we added the algorithm we recommend for ALK positive NSCLC treatment in Page16 lines 949-965 and Figure 2, based on the statistics form previous studies and the results of studies on resistance mechanisms.

Point 4: There were no figures. The authors should summary the mechanisms as a figure and add a timeline figure of key findings according years.

Response 4: As reviewer suggested, we provided a table of "Timeline of discovery of EML4-ALK and US FDA approval of ALK TKIs" as Figure 1.

Point 5: The authors describe each drug. What was the mechanisms of these drugs? Can they be categorized?

Response 5: We noted that since lorlatinib is a macrocyclic TKI, it is smaller and more compact than acyclic TKIs such as 1st/2nd generation TKIs, and for this reason it is sensitive to ALK resistance mutations such as G1202R on Page7 line 480-482. We have also described it as a characteristic of molecule in Table 4.

Reviewer 2 Report

This is an interesting and comprehensive overview on ALK re-arranged NSCLC. All major aspects have been considered. However, the data is rather overwhelming and not easy to review. Hence, I would recommend to re-structure large parts: For example, when summarizing the respective agents, most data could be put in tables. In particular for long chapters, there should be a brief conclusion to guide the reader to the most essential findings.

It is difficult to compare certain aspects. Hence, tables such as tables 1-3 are really helpful. There should be a paragraph on the different side effects that re quite different, indeed. Here, a sentence how to take these drugs (with/without food, how many tables is the normal dose et) would be helpful.

For chapters 3-5: A brief rationale why these drugs should be combined would be helpful. In chapter 5: I would focus on ALK only. Data on EGFR etc is rather confusing and would need a whole different further discussion.

Table1: Maturity is missing which is somewhat important, e.g. medium/max follow-up

Chapter 7 is definitively important but also lacks a structure to grasp the different resistance profiles. Moreover, the data on ALK TKI sequencing is missing (the only aspect in this article) but this is the most clinically important! I would shorten chapter 8 und move it to a much earlier position, eg straight after introduction with the most important aspects of diagnosis (IH versus FISH versus PCR versus NGS etc)

Author Response

Response to Reviewer 2 Comments
We thank the reviewer for insightful comments, which we feel have helped us to improve our manuscript. Our specific responses to the points raised are as follows:

Point 1: There should be a paragraph on the different side effects that re quite different, indeed. Here, a sentence how to take these drugs (with/without food, how many tables is the normal dose et) would be helpful.

Response 1: As reviewer suggested, we added dosage and common/typical adverse effects in Table 4.

Point 2: For chapters 3-5: A brief rationale why these drugs should be combined would be helpful.

Response 2: We added a new chapter 2, so chapters 3-5 were changed to chapters 4-6. In chapter 4, we added the mechanism for cancer of angiogenesis ihbibitors in P12 line 691-692. In line691-693, we described that since combination therapy with angiogenesis inhibitors and TKIs has been shown to be effective for EGFR-muted NSCLC, it is expected to be similarly effective for ALK-rearranged NSCLC. We have added to chapter 5 of Page12 line 718-721 that
the presence of ALK mutation is associated with PD-L1 expression and therefore immune checkpoint inhibitors are expected for ALK-rearranged NSCL. Chapter 6 shows that chemotherapy, including pemetrexed, has been shown to be effective in ALK-positive NSCLC in previous clinical studies.

Point 3:In chapter 5: I would focus on ALK only. Data on EGFR etc is rather confusing and would need a whole different further discussion.

Response 3: As reviewer suggested, because the data on EGFR is confusing, we summarized the data of NEJ009 and another phase3 trial briefly. The only clinical trial on combination therapy of platinum doublet and ALK-TKI is the phase II trial in Japan which is ongoing. Previous clinical studies have shown that pemetrexed is effective in ALK-rearranged NSCLC. We noted that if the combination of pemetrexed and ALK-TKI is well tolerated, it may improve PFS and OS on page14 line 818-819.

Point 4: Table1: Maturity is missing which is somewhat important, e.g. medium/max follow-up

Response 4: We agree with reviewer’s suggestion and we added median follow-up duration for PFS/OS to Table1.

Point 5: Chapter 7 is definitively important but also lacks a structure to grasp the different resistance profiles. Moreover, the data on ALK TKI sequencing is missing (the only aspect in this article) but this is the most clinically important!

Response 5: We agree with reviewer’s suggestion. We classified Chapter 8 into 8.1 Primary resistance mechanisms, 8.2 Acquired resistance mechanisms, 8.2.1 ALK-dependent resistance mechanisms, and 8.2.2 ALK-independent resistance mechanisms. In addition, we added treatment algorithm for ALK-rearranged advanced NSCLC as 8.3.

Point 5: I would shorten chapter 8 und move it to a much earlier position, eg straight after introduction with the most important aspects of diagnosis (IH versus FISH versus PCR versus NGS etc)

Response 5: As reviewer suggested, we have added descriptions of FISH, IHC, and NGS to the previous chapter 8, and recreated it as a new chapter 2 “Diagnosis”.

Reviewer 3 Report

Reviewer’s Comment:

The current review on therapeutic strategies for anaplastic-lymphoma-kinase-rearranged (ALK) non-small cell lung cancer by Fukui T et al., comprehensively demonstrated the past, present, and future perspectives of ALK-rearranged NSCLCs treatment, covering recent clinical trials with ALK-TKIs combining angiogenesis and immune checkpoint inhibitors. This intricate review pretty much covers everything except one minor point.

The introduction section is small, nevertheless, authors are highly encouraged to incorporate a decent paragraph at the end of the introduction, covering background for the disease-development, the basic etiology of ALK-rearranged NSCLCs, targeting the general readers whereas, sharing the basic knowledge about this subtype of lung cancer.

Author Response

Response to Reviewer 3 Comments
We thank the reviewer for insightful comments, which we feel have helped us to improve our manuscript. Our specific responses to the points raised are as follows:

Point 1: The introduction section is small, nevertheless, authors are highly encouraged to incorporate a decent paragraph at the end of the introduction, covering background for the disease-development, the basic etiology of ALK-rearranged NSCLCs, targeting the general readers whereas, sharing the basic knowledge about this subtype of lung cancer.

Response 1:In Page2 line 39-41, we added the history of the discovery of ALK mutation and that ALK is considered to be involved in the development of the central nervous system. In  Page2 line 48-50 and Fig1, we added the timeline of the invention of ALK-TKI.

Reviewer 4 Report

The manuscript by Fukui and collaborators summarizes the current knowledge regarding therapy for ALK-rearranged advanced NSCLC. The authors summarized in brief the recent clinical trials of ALK tyrosine kinase inhibitors, and combination therapy with angiogenesis inhibitors and immune checkpoint inhibitors as a treatment for ALK-rearranged NSCLC.

Overall, the manuscript is well-structured, provides a good coverage of the topic, and can be useful to researchers working in this area.

However, there are several points that should be addressed to improve the paper. The introduction is too brief and should be expanded to provide more information regarding the ALK oncogene. The use of immune checkpoint inhibitors and angiogenesis inhibitors in ALK-rearranged NSCLC treatment as monotherapy is also not properly introduced. Furthermore, the authors later discuss the use of chemotheraputics for ALK-rearranged NSCLC but do not provide any information in the Introduction section.

A Table summarizing the clinical trials involving combination therapy with ALK tyrosine kinase inhibitors and immune checkpoint inhibitors would also be useful.

Author Response

Response to Reviewer 4 Comments
We thank the reviewer for insightful comments, which we feel have helped us to improve our manuscript. Our specific responses to the points raised are as follows:

Point 1:The introduction is too brief and should be expanded to provide more information regarding the ALK oncogene.

Response 1: In Page2 line 39-41, we added the history of the discovery of ALK mutation and that ALK is considered to be involved in the development of the central nervous system.

Point 2:The use of immune checkpoint inhibitors and angiogenesis inhibitors in ALK-rearranged NSCLC treatment as monotherapy is also not properly introduced.

Response 2: We introduce two retrospective studies of IO monotherapy on Page 12, lines 726-732. We have found no trials evaluating monotherapy with angiogenesis inhibitors for ALK-rearranged NSCLC.

Point 3:Furthermore, the authors later discuss the use of chemotheraputics for ALK-rearranged NSCLC but do not provide any information in the Introduction section.

Response 3: I added a description about chemotherapy in the introduction section of lines 53 and 55 on page 2.

Point 4: A Table summarizing the clinical trials involving combination therapy with ALK tyrosine kinase inhibitors and immune checkpoint inhibitors would also be useful.

Response 4: We have created Table 5 to summarize recent and ongoing clinical trials including combination therapy with ALK tyrosine kinase inhibitors and immune checkpoint inhibitors.

Round 2

Reviewer 1 Report

The authors have answered my questions.

Author Response

Thank you for taking the time to read this manuscript. Your advice has helped us to write the better review. We really appreciate it.

Reviewer 2 Report

Minor comments:

  • I would leave ImPower150 out here since it is only 21 patients and some of those were false positive. To my knowledge there is no publication reporting on ALK+ only of this trial. Hence: delete
  • Figure 2: Pls state Met inhibitor instead of Crizotinib sincere there are several TKIs (and several with superior to crizotinib)

Reviewer 3 Report

"ALK was first discovered as a fusion partner in the t(2;5) chromosomal in anaplastic large cell lymphoma in 1994 by Morris et al" - the first sentence in the introduction section.

  1. What does it mean by t(2;5) chromosome?
  2. The time of discovery of ALK, i.e., 1994 is not matching with the time-line mentioned in Fig.1.

Reviewer 4 Report

The authors have made a poor effort to address my previous concerns. The manuscript in its current form is not suitable for publication.

Round 3

Reviewer 4 Report

The authors have significantly improved the manuscript.